# Picosecond pulse-shaping for strong three-dimensional field-free alignment of generic asymmetric-top molecules

Terry Mullins [1], Evangelos T. Karamatskos[1,2], Joss Wiese [1,3,4], Jolijn Onvlee [1,4,6], Arnaud Rouzée[5], Andrey Yachmenev [1,4], Sebastian Trippel[1,4] & Jochen Küpper [1,2,3,4 ✉]

Fixing molecules in space is a crucial step for the imaging of molecular structure and dynamics. Here, we demonstrate three-dimensional (3D) field-free alignment of the proto-typical asymmetric top molecule indole using elliptically polarized, shaped, off-resonant laser pulses. A truncated laser pulse is produced using a combination of extreme linear chirping and controlled phase and amplitude shaping using a spatial-light-modulator (SLM) based pulse shaper of a broadband laser pulse. The angular confinement is detected through velocity-map imaging of $H^+$ and $C^{2+}$ fragments resulting from strong-field ionization and Coulomb explosion of the aligned molecules by intense femtosecond laser pulses. The achieved three-dimensional alignment is characterized by comparing the result of ion-velocity-map measurements for different alignment directions and for different times during and after the alignment laser pulse to accurate computational results. The achieved strong three-dimensional field-free alignment of $\langle \cos^2\delta \rangle = 0.89$ demonstrates the feasibility of both, strong three-dimensional alignment of generic complex molecules and its quantitative characterization.

[1] Center for Free-Electron Laser Science, Deutsches Elektronen-Synchrotron DESY, Notkestraße 85, 22607 Hamburg, Germany. [2] Department of Physics, Universität Hamburg, Luruper Chaussee 149, 22761 Hamburg, Germany. [3] Department of Chemistry, Universität Hamburg, Martin-Luther-King-Platz 6, 20146 Hamburg, Germany. [4] Center for Ultrafast Imaging, Universität of Hamburg, Luruper Chaussee 149, 22761 Hamburg, Germany. [5] Max Born Institute, Max-Born-Straße 2a, 12489 Berlin, Germany. [6] Present address: Institute for Molecules and Materials, Radboud University, Heijendaalseweg 135, 6525 AJ Nijmegen, The Netherlands. ✉email: jochen.kuepper@cfel.de

Laser-induced alignment of gas-phase molecules has proven to be an efficient way to access the molecular frame[1–3]. It was extensively used in high-harmonic-generation-spectroscopy[4,5], strong-field-ionization[6–8], x-ray-diffraction[9,10], and electron-diffraction[11–15] experiments, enabling the imaging of molecular structure and dynamics directly in the molecular frame. Furthermore, it was crucial for retrieving the shapes of molecular orbitals[16–18].

Such advanced imaging technologies are especially important for complex molecules, i.e., asymmetric tops without any rotational symmetry, which is the case for almost all molecules on earth. Thus it is of utmost importance to develop laser alignment into a practical tool for such molecules. This would, for instance, maximize the information content of atomic-resolution imaging experiments[19,20], as already suggested for the coherent diffractive x-ray imaging of biological macromolecules more than 15 years ago[21]. In order to minimize perturbations by external fields, this should be achieved in a laser-field-free environment. The associated problems are twofold: the rotational dynamics of these (generic) asymmetric top molecule molecules are very complicated and incommensurate[22,23]. Moreover, the standard approaches to characterize the 3D degree of alignment using ion imaging of atomic fragments, mostly halogen atoms, recoiling along a well-defined molecular axis, do not work.

One-dimensional alignment of linear and (near) symmetric top molecules has been demonstrated extensively and really pushed to the limits[1,2,17,24–27], including concepts for time-domain detection methods for asymmetric top molecules[28]. Furthermore, the three-dimensional (3D) control of rotation-symmetric molecules, typically during long laser pulses, was demonstrated by multiple groups, making use of highly polarizable halogen atoms for large polarizability effects as well as their symmetric fragmentation dynamics for characterization[17,29–34]. This was extended to the field-free 3D alignment of asymmetric top molecules using sequences of either orthogonally polarized[35,36] or elliptically polarized laser pulses[37], as well as long-lasting field-free alignment in helium nanodroplets[38] using rapidly truncated pulses and the alignment of one (generic) asymmetric top molecule 6-chloropyridazine-3-carbonitrile using long laser pulses[23,39].

Here, we demonstrate and characterize the strong laser-field-free 3D alignment of the prototypical (bio)molecule indole ($C_8H_7N$, Fig. 1a), a good representative of the general class of molecules without any rotational symmetry and without any good leaving-group fragments for standard characterization. We use a combination of a shaped, truncated, elliptically polarized laser pulse with a short kick pulse before truncation to induce strong 3D alignment. The degree of alignment is characterized through strong-field multiple ionization and subsequent velocity-map imaging (VMI) of $H^+$, $C^+$, $C^{2+}$, and $CH_xN^+(x = 0, 1, 2)$ fragments, combined with computational results to disentangle the temporal and angular dependence of the alignment. Our approach shows that the molecular frame even of generic asymmetric top molecules can be accessed.

## Results

**Experimental setup.** The experimental setup was described elsewhere[40]. Briefly, molecules were cooled in a supersonic expansion from a pulsed Even–Lavie valve[41], operated at a temperature of 80°C and at a repetition rate of 100 Hz. Around 1.4 mbar of indole was seeded in 95 bar of He, which was expanded into vacuum. The lowest-energy rotational states were selected using an electrostatic deflector[42,43]. Inside a VMI spectrometer, the strongly deflected molecules were aligned using a shaped 250-ps long laser pulse with a peak intensity of

$\sim 1.25 \times 10^{12}$ W/cm². These pulses were produced by a commercial laser system (Coherent Legend Elite Duo) with a 1 kHz repetition rate and a spectrum similar to a rounded saw tooth. The pulse was strongly negatively chirped to a duration of ~600 ps using a grating-based compressor[40] before further shaping. The alignment laser pulses were elliptically polarized with a 3:1 intensity ratio between major and minor axes.

The strongly chirped 9 mJ pulses were sent through a zero dispersion 4f pulse shaper[44] with a spatial light modulator (SLM, Jenoptik S640d) situated at the Fourier plane in order to generate a truncated pulse with a fast fall-off time. The most relevant part of the shaped temporal intensity profile, around the cutoff, is shown in Fig. 2a. The pulse consisted of a slow rise beginning at −250 ps (not shown), followed by some amplitude modulation, a short kick with a duration of 2.6 ps (FWHM), and, finally, a fast truncation. The SLM was specifically used for spectral phase modulation of spectral components between 815 and 816 nm. In addition, wavelengths longer than ~816 nm were blocked by a razor blade, situated directly in front of the SLM. This was necessary due to Nyquist-sampling limits encountered. We note that the combination of phase shaping and spectral truncation with the razor blade improved the temporal fall-off time by a factor of 2.5 down to 3.3 ps, i.e., to within the noise level of the measurement, which was below 1% of the signal peak, compared to simply cutting the spectrum[34].

The post-pulse observed at ~13 ps is unwanted and probably originates from imperfect phase compensation from the SLM or space-time coupling in the pulse-shaping setup. However, the post-pulse is irrelevant to the degree of alignment within the first 10 ps after the temporal truncation, which corresponds to the important temporal region investigated in this experiment.

A second, time-delayed, laser pulse with a pulse duration of 35 fs (FWHM) and a peak intensity of $4.6 \times 10^{14}$ W/cm² was used to multiply ionize indole, resulting in Coulomb explosion. These pulses were circularly polarized to avoid any secondary dynamics induced by electron rescattering[45] and in order to minimize the bias from geometric alignment.

Velocity-mapped fragments were detected on a microchannel plate (MCP) detector equipped with a phosphor screen. The voltage on the MCP was switched between 2050 V (MCP "on") and 1150 V (MCP "off") using a fast switch (Behlke HTS 31-03-GSM) with 100 ns rise- and fall-times to select the different ion fragments based on their times of flight. A camera (Optronis CL600) recorded single-shot images of the phosphor screen at 200 Hz. Images without pulses from the molecular beam were subtracted from those with the molecular beam to account for any signal from background molecules in the interaction region. After the selection of a suitable two-dimensional (2D) radial range, the degree of alignment $<\cos^2 \theta_{2D}>$ was computed. Schematic visualization of the imaging geometry is shown in Fig. 1c, with the detector plane defining the $(Y, Z)$ plane. The angle $\theta_{2D}$ is defined as the polar angle in the detector plane with respect to the Z axis. The angle $\alpha$ defines the orientation of the major polarization axis of the alignment laser ellipse with respect to the lab-frame Z axis, where $\alpha = 0°$ stands for parallel alignment and $\alpha = 90°$ for perpendicular alignment, see Fig. 1c.

**Ion-momentum distributions.** The in-plane principal axes of inertia $(a, b)$ and polarizability $(z_I, x_I)$ are shown in the ball-and-stick representation of indole in Fig. 1a. Both axis frames lie in the plane of the molecule with an angle of 2.75° between them, whereas the $c$ and $y_I$ axes are perpendicular to that plane. The alignment process fixed the $z_I$ and $x_I$ axes in the laboratory frame but not their directions, leading to four simultaneously present orientations of the molecule. Upon Coulomb explosion, several

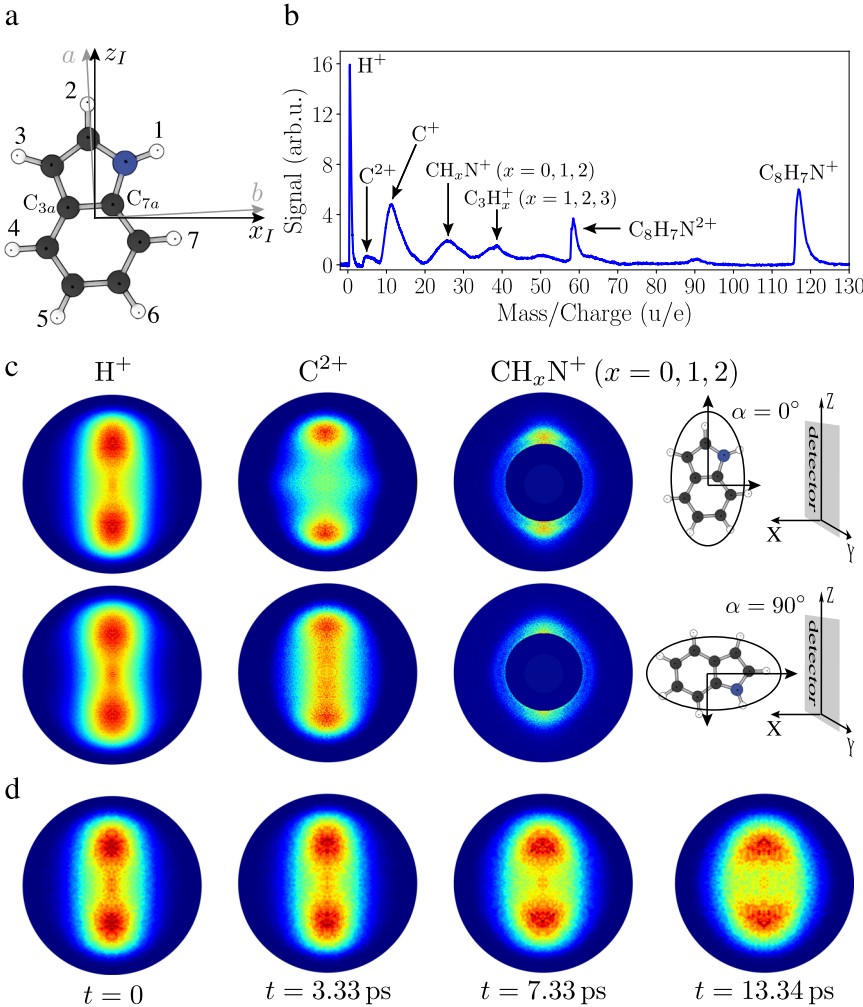

**Fig. 1 Molecular structure, mass spectrum, and ion images of indole. a** The structure of the indole molecule with its principal axes of inertia and polarizability, labeled by $a$, $b$, $c$ and $x_I$, $y_I$, $z_I$ ($\alpha_{y_I} < \alpha_{x_I} < \alpha_{z_I}$), respectively. **b** TOF mass spectrum of indole. **c** 2D momentum distributions for $H^+$, $C^{2+}$ and $CH_xN^+ (x = 0, 1, 2)$ fragments at peak alignment at $t = 3.3$ ps, with the major axis of the alignment laser polarization parallel (first row, $\alpha = 0°$) or perpendicular (second row, $\alpha = 90°$) to the detector plane. **d** Time-snapshots of 2D momentum distributions of $H^+$ fragments for the case of parallel laser polarization.

fragmentation channels were detected. The resulting time-of-flight mass spectrum is depicted in Fig. 1b.

Several ionic fragments showed anisotropic momentum distributions. The ion-momentum distributions of $H^+$, $C^{2+}$ and $CH_xN^+ (x = 0, 1, 2)$ for a delay time of $t = 3.3$ ps, corresponding to the highest observed degree of alignment, are shown in Fig. 1c for two orientations of the alignment laser, i.e., with the main polarization axis being parallel, $\alpha = 0°$, or perpendicular, $\alpha = 90°$, to the plane of the detector. $t = 0$ corresponds to the peak intensity of the alignment laser field. Furthermore, ion-momentum distributions of $H^+$ for time delays of $t = 0$, 3.3, 7.3, and 13.3 ps are shown in Fig. 1d. The strongest field-free alignment was observed near $t = 3.3$ ps. At later delay times, the dephasing of the rotational wavepacket leads to a decrease of the molecular alignment, as seen in the momentum distributions recorded at time delays of $t = 7.3$ ps and $t = 13.3$ ps. Momentum distributions of other fragments displaying alignment are shown in Supplementary Note 1.

Indole does not contain unique markers, like halogen atoms, which would allow us to easily experimentally access the degree of alignment. Therefore, all ions with a given mass to charge ratio, produced through multiple ionization with subsequent Coulomb explosion, potentially contributed to the measured 2D momentum distributions. There are seven sites in the indole molecule

from which the $H^+$ fragments originate and eight sites for the $C^{2+}$ fragments, see Fig. 1a. Each molecular site will result in different momentum and recoil axis of the ionic fragment, and the total measured distribution is the sum of all of them.

The delay-dependent measured 2D degree of alignment is shown for a variety of fragments in Fig. 2. Assuming axial recoil, $H^+$ fragments would have measurable momentum components only within the $ab$ plane of indole[46]. Hence, the $H^+$ fragments are a priori a good measure of the planar alignment of indole in the laboratory frame. The slow rise of the alignment pulse confined the plane of the indole molecules in a quasi-adiabatic fashion[25,40] to a measured maximum degree of alignment of $<\cos^2 \theta_{2D}>_{H^+}^{exp} = 0.72$. Following the kick at the end of the alignment pulse, the degree of alignment increased slightly to $<\cos^2 \theta_{2D}>_{H^+}^{exp} = 0.73$ before monotonically decreasing over ~10 ps to $<\cos^2 \theta_{2D}>_{H^+}^{exp} = 0.62$. The permanent alignment of $<\cos^2 \theta_{2D}>_{H^+}^{exp} = 0.62$ was slightly higher than the value $<\cos^2 \theta_{2D}>_{H^+}^{exp} = 0.60$ observed without alignment laser; the latter is due to the geometric alignment from an isotropic distribution. At a delay of 3.3 ps the intensity of the alignment pulse decreased to 1% of its maximum, and the "field-free" region began. At this delay the degree of alignment was $<\cos^2 \theta_{2D}>_{H^+}^{exp} = 0.73$, which was even larger than the alignment measured just before the kick, confirming that the planar alignment in the field-free region

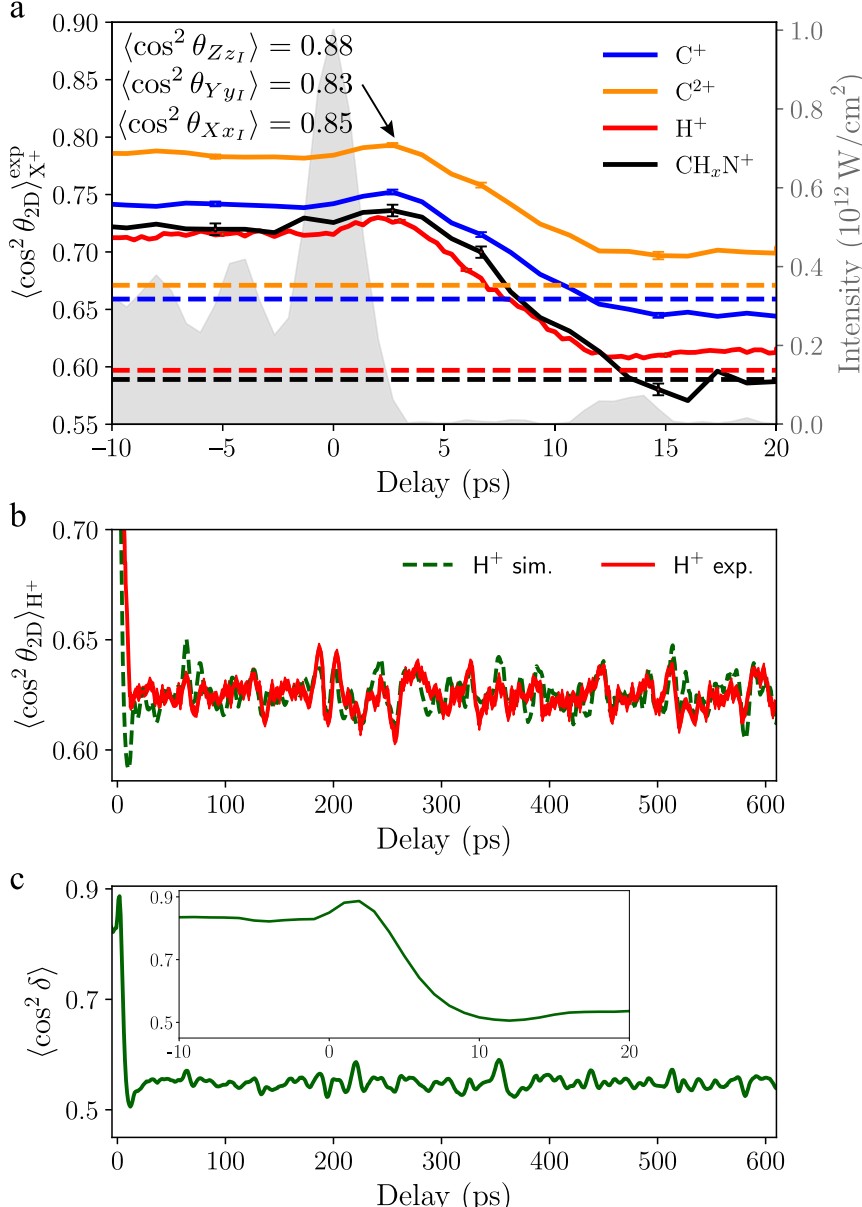

**Fig. 2 Temporal evolution of the alignment of indole for the parallel alignment geometry ($\alpha = 0°$). a** The solid lines show the measured 2D degree of alignment $\langle\cos^2\theta_{2D}\rangle^{exp}_{X^+}$ for different fragments $X^+$ and the dashed lines indicate values of the 2D degree of alignment obtained without alignment laser. Statistical error bars, representing the standard error, are shown for selected delays. The grey area shows the intensity profile of the alignment laser pulse. Also shown are the three expectation values $\langle\cos^2\theta_{Xx_I}\rangle$, $\langle\cos^2\theta_{Yy_I}\rangle$ and $\langle\cos^2\theta_{Zz_I}\rangle$, with $\theta_{lj}$ being the angles formed between the three main polarizability axes $j \in (x_I, y_I, z_I)$ of indole and the three lab-frame axes $l \in (X, Y, Z)$, and their computed values for the actual 3D alignment of indole. **b** The alignment revival structure of $H^+$ fragments for longer times is shown in red, with the line thickness corresponding to the experimental standard error of the measurements. The dotted green line shows the fitted simulation for the $H^+$ fragment. **c** Simulated 3D degree of alignment, characterized through the single-scalar metric $\langle\cos^2\delta\rangle$, see text/SI for details. Note the peak after truncation reaching $\langle\cos^2\delta\rangle = 0.89$ at $t = 3.3$ ps.

was even better than for an adiabatic alignment pulse[47,48]. All other fragments showed similar distributions to the $H^+$ fragment, with the measured maximum degree of alignment being largest for the $C^{2+}$ fragment. The differences in the measured alignment between the $H^+$, $C^+$, $C^{2+}$ and $CH_xN^+(x = 0, 1, 2)$ fragments can be attributed to non-axial recoil or to the geometry of Coulomb explosion fragmentation, i.e., the velocity vectors of the fragments in the molecular frame.

**Characterization of 3D alignment.** To determine the 3D alignment of indole, an additional observable is required that

characterizes the in-plane alignment, i.e., the alignment of the most polarizable axis of indole $z_I$ with respect to the main polarization axis of the alignment field. This information can be accessed by measuring the angular distribution of the ionic fragments within the indole plane. By rotating the polarization ellipse of the alignment laser around the laser propagation axis at a fixed delay time of 3.3 ps, the laboratory axes to which the $a$ and $b$ axes of indole align were commensurately rotated. In the laboratory frame, the transverse momenta of ionic fragments recoiling within the plane of indole will depend on the ellipse-rotation angle $\alpha$, between $\alpha = 0°$ for parallel and $\alpha = 90°$ for perpendicular orientation, see Fig. 1c. By counting only those

fragments impinging at the center of the detector, within a small radius of 20 pixels, the distribution of fragments within the plane can be determined[49]. Note that the size of the VMI images, as shown in Fig. 1c, d, was $480 \times 480$ pixels. Full tomographic measurements were carried out for $H^+$, $C^+$, $C^{2+}$ and $CH_xN^+$ ($x = 0, 1, 2$) at $t = 3.3$ ps for $\alpha = 0°$–$180°$($\Delta\alpha = 2°$). A visualization of the 3D reconstructed signal is shown for the $H^+$ fragment in Supplementary Note 2. For both, the $H^+$ and the $C^{2+}$ fragments, the 3D velocity distributions were quasitoroidal, i.e., no considerable density at or around the origin was observed. The signals, measured at the center of the VMI in the 2D data, can thus be attributed to in-plane fragments recoiling along the detector normal, proving the validity of the approach chosen. The approach itself is equivalent to using narrow slices through the fully reconstructed 3D momentum distributions from tomographic measurements; however, it presents certain advantages. These are in particular a decreased data acquisition time, since less data for the characterization of the angular distributions are required than for a full tomographic reconstruction, and finally the actual tomographic reconstruction of the 3D momentum distributions can be circumvented, rendering the chosen approach more practical.

Angular scans of this "masked VMI" are shown in Fig. 3 for $H^+$ and $C^{2+}$. For both fragments, we observed a clear angle-dependent structure on top of a significant isotropic background. $C^{2+}$ ions show two smaller peaks at $\alpha \approx \pm30°$ and a much stronger peak at $\alpha \approx 90°$. The $H^+$ signal shows a peak at $\alpha \approx 90°$, similar to $C^{2+}$, and a smaller peak at $\alpha \approx 0°$. Note that at $90°$, the alignment laser's major polarization axis is pointing along the detector normal. Direct extraction of the in-plane degree of alignment from the experimentally obtained in-plane angular distribution was not possible due to a large isotropic background. Furthermore, the degree of molecular alignment retrieved from the angular momentum distributions of $H^+$ and $C^{2+}$ can be misrepresented mainly due to two reasons—the many-body Coulomb breakup of the multiply charged indole cation, with ionic fragments violating the axial-recoil approximation, as well as the indistinguishability of fragments emitted at different molecular sites.

**Computational calibration of the degree of alignment.** In order to determine the actual 3D degree of alignment, we performed comprehensive variational simulations of the rotational dynamics of indole in the presence of the alignment field. We employed the general variational approach RichMol[50,51] to compute time-dependent rotational probability density distributions for different delay times. In order to incorporate the experimental conditions and to achieve better agreement, we took into account the non-thermal distribution of rotational states in the deflected part of the molecular beam and laser focal-volume averaging.

The total probability density distributions of $C^{2+}$ and $H^+$ were modeled as the weighted sums of contributions from the individual atoms. As a consequence of orientational averaging, most of the arising ions do not possess a unique recoil direction within the polarization frame of the alignment laser. We accounted for this by using equal weights for pairs of atoms H1 and H3, H4 and H7, H5 and H6, C3a and C7a, C4 and C7, and C5 and C6. As the recoil axes, we choose vectors connecting carbon atoms to the center of mass of the molecule for $C^{2+}$ and vectors along molecular C–H and N–H bonds for $H^+$. To reproduce the experimental data the simple axial recoil approximation yielded excellent agreement for $C^{2+}$, whereas for $H^+$ we had to account for non-axial recoil by convoluting the calculated probability density distributions of hydrogen atoms with a Gaussian function of a solid angle representing angular displacement from the recoil vector. The weights and the FWHM parameter of this Gaussian function were determined in a least-squares fitting procedure to the measured alignment revival trace and the angle-dependent masked VMI data. The obtained parameters are specified in Supplementary Note 3. The results of the fit show very good agreement with the experimental alignment revivals in Fig. 2b and excellent agreement with the integrated in-plane angle-dependent projections through the 3D momentum distribution in Fig. 3.

This excellent agreement confirms the correct representation of the experiment by our quantum simulations. In principle, experimental input parameters to the simulations could be varied, but this was not necessary due to an accurate determination of experimental parameters. Such a fitting procedure is also undesirable, due to the time-consuming nature of the simulations. The obtained planar alignment in terms of squared direction cosines[52] at $t = 3.3$ ps is $<\cos^2\theta_{Y_{y_I}}> = 0.83$ and $<\cos^2\theta_{X_{x_I}}> = 0.85$. These values are higher than the measured values, which is due to non-axial recoil of the $H^+$ fragments and different recoil axes contributing to the measured ion-momentum distributions. The computed in-plane degree of alignment at $t = 3.3$ ps is $<\cos^2\theta_{Z_{z_I}}> = 0.88$, which we also assign as the experimental value due to the excellent match of the angular distributions in Fig. 3. Simulated time-dependent alignment revivals can be found in Supplementary Note 3. A single-scalar metric describing the overall degree of 3D alignment $\cos^2\delta = \frac{1}{4}(1 + \cos^2\theta_{Z_{z_I}} + \cos^2\theta_{Y_{y_I}} + \cos^2\theta_{X_{x_I}})$[52] is shown in Fig. 2c. A maximum degree of field-free alignment of $<\cos^2\delta> = 0.89$ was obtained, which is comparable to or even larger than one can achieve for complex asymmetric top molecules using adiabatic alignment techniques[39,53] and clearly sufficient for molecular-frame coherent diffractive imaging[19,20].

## Discussion
We demonstrated strong laser-field-free 3D alignment of the prototypical complex (generic) asymmetric top molecule indole induced by shaped truncated quasi-adiabatic laser pulses. Both the amplitude and the phase of a strongly chirped broadband alignment laser pulse were tailored using an SLM, which allowed us to produce very short truncation times, unachievable with amplitude truncation alone. The combination of quasi-adiabatic alignment with a kick pulse directly before the sudden truncation produced a higher degree of alignment under field-free conditions than in the field. The already achieved strong degree of alignment is limited by the initially populated states in the molecular beam[54]

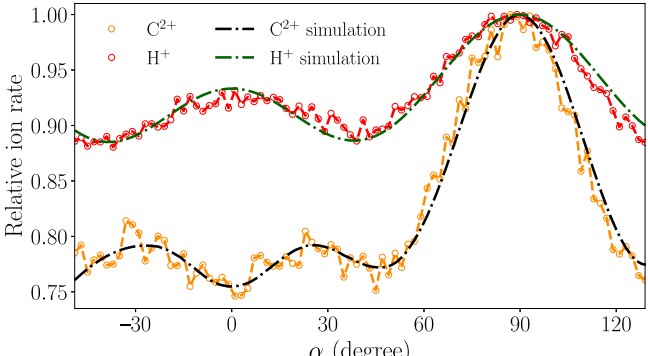

**Fig. 3 In-plane angle-dependent ion count rates.** Measured masked VMI ion count rates of $H^+$ and $C^{2+}$ fragments as the major axis of the alignment laser's polarization ellipse is rotated; see text for details. Circles (red and orange) indicate measured values, the dashed lines (green and black) are simulations based on fitted atomic-ion contributions, see text for details.

and could be further improved through even colder molecular beams[42,43].

We have developed and tested a versatile approach to characterize molecular alignment in 3D by performing separate measurements of planar alignment and in-plane tomography. Planar alignment was characterized as the time-dependent alignment trace of the $H^+$ fragments. In-plane alignment was characterized using the angular dependence of $H^+$ and $C^{2+}$ fragment distributions at the center of the detector, obtained by rotating the laser polarization ellipse and thus the molecule in the plane perpendicular to the detector. Robust variational simulations of the alignment dynamics of indole, considering weighted contributions of fragments emitted non-axially at different molecular sites, reproduced the experiment with high accuracy.

This demonstration of strong field-free alignment for an asymmetric top rotor without rotational symmetries and without any good ionic fragments for the characterization of the alignment paves the way for strong field-free alignment of any arbitrary molecule. This opens up important prospects for probing native (bio)molecules in the molecular frame[20,21,55] without chemically attaching marker atoms that influence the function and properties of the molecule.

## Methods
Our general experimental setup was described previously[40,43] and the specific details for the current experiment were provided in the main text. Software used for the simulations were described elsewhere[50,51] and specific details were described in the main text and the Supplementary Methods.

## Data availability
The data that support the findings of this study are available from the repository at https://doi.org/10.5281/zenodo.5897172.

## Code availability
Quantum rotational dynamics simulations were performed using Richmol, available at https://github.com/CFEL-CMI/richmol. Further codes used for the analysis of experimental data and analysis are available at https://doi.org/10.5281/zenodo.5897172.

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

## Acknowledgements

We thank Stefanie Kerbstadt for helpful discussions. We acknowledge support by Deutsches Elektronen-Synchrotron DESY, a member of the Helmholtz Association (HGF), and the use of the Maxwell computational resources operated at Deutsches Elektronen-Synchrotron DESY. This work has been supported by the Deutsche Forschungsgemeinschaft (DFG) through the priority program "Quantum Dynamics in Tailored Intense Fields" (QUTIF, SPP1840, KU 1527/3, AR 4577/4, YA 610/1; J.K., A.R., A.Y.) and by the Clusters of Excellence "Center for Ultrafast Imaging" (CUI, EXC 1074, ID 194651731; J.K.) and "Advanced Imaging of Matter" (AIM, EXC 2056, ID 390715994, J.K.), and by the European Research Council under the European Union's Seventh Framework Programme (FP7/2007-2013) through the Consolidator Grant COMOTION (614507; J.K.). J.O. gratefully acknowledges a fellowship of the Alexander von Humboldt Foundation.

## Author contributions

J.K. and A.R. devised the study. T.M., J.W., J.O., and S.T. set up the experiment, and T.M., J.W., and J.O. carried out the experiment. A.Y. and E.T.K. developed the simulations and performed the calculations. T.M., J.W., J.O., and E.T.K. analyzed the experimental and computational data. J.K. supervised the study. All authors were involved in interpreting the data and clarifying concepts in the experiment and its analysis. T.M. and E.T.K. wrote the initial manuscript and all authors were involved in discussing and editing the manuscript.

## Funding

## Competing interests

The authors declare no competing interests.
