## [Peer Review File · Nature Communications]

Picosecond pulse-shaping for strong three-dimensional field-free alignment of generic asymmetric-top moleculesREVIEWER COMMENTS

Reviewer #1 (Remarks to the Author):

The authors claim a impressive degree of field-free 3D alignment of a somewhat large planar molecule using a long pulse that has been truncated and shaped to kick the molecules into better alignment just as the pulse intensity decays to a negligible level. The demonstration of this idea is an important step forward for the use of aligned molecules in experiments seeking to make “molecular movies”. The molecule indole also has no convenient alignment markers for momentum spectroscopy, so the measurements of 3D alignment is particularly challenging. But I have several questions and concerns, which I have listed below.

1. Fig. 2 shows the temporal shape of the pump laser with an intensity scale. How was the peak intensity of the pulse (at 0 ps) estimated? How large is the error bar on this measurement? Was the possible error in the intensity considered in the fitting of the delay-dependent data?? Since the determination of the expectation values of the various squared-cosines relies heavily on simulations, it is important to understand error in the estimated intensity and the effect of the error in the peak intensity.

2. The discussion of probe-selectivity, or geometric alignment, is limited. Fig. 2 shows that the measured fragment all have significantly aligned momentum distributions without the pump pulse, but there is no mention of geometric alignment in the discussion of the simulations. Is geometric alignment accounted for? If yes, how? If not, why not? Do the simulations accurately reproduce the geometric alignment seen in Fig. 2?

There is also no mention of the probe intensity or pulse duration in the manuscript.

3. The use of fragments near-zero momentum in the detector plane is well-justified for H⁺ fragment through the tomographic measurement of the 3D momentum measurement. But no such measurement is discussed for the C₂⁺ fragment. Was such a measurement made?

And for H⁺, doesn't the tomography data contain the data that is shown in Fig. 3? Wouldn't a narrow cut through the “doughnut” shown in Fig.3 of SI reproduce the plot for H⁺ shown in Fig. 3? If tomography is necessary to verify that the “masked VMI” measurements are useful – and I think it is – I'm not sure I see the point in using the 2D VMI data directly.

Using tomography to characterize 3D alignment isn't a new approach as claimed by the authors. See Ren et al. Phys. Rev. Lett. 112, 173602 (2014).

Visually, the images showing the 3D momentum distributions of H⁺ in the SI is unsatisfactory. The manuscript relies heavily on fitting data to simulations, but the visual representation of the data is such that details of the distribution are hard to see. I suggest the authors consider a better-looking representation, perhaps showing cuts through the distribution in addition to the full 3D distribution.

4. That brings me to what's my most important concern about this manuscript. The simulation of the momentum distribution seems to have a lot of free parameters to describe data that has little structure. There are the weights for each fragment, the Gaussian function that accounts for non-axial recoil of H⁺ (this function and its parameters are not provided in the SI, although the main article says it is), any parameters needed to account for geometric alignment (if it is accounted for at all – I suspect it is not since the probabilities for various fragments are numbers rather than angle-dependent functions).

But the pump pulse parameters are not varied. The fitting procedure doesn't really retrieve any alignment parameters – it just serves to confirm that the data agrees with the rigid-rotor TDSE simulations after the simulation data for molecular axis distributions have been suitably fudged. The variational calculation only determines the fudging parameters required to get agreement. Are these parameters reasonable? Is the model used for the fudging appropriate and not unnecessarily large? How are we to judge? On the one hand the data in Fig 2(b) is rich in structure, but the fit is not particularly convincing (and I'd like to see the fit to the data in Fig. 2(a)). On the other, the fit to data in Fig. 3 is good but these two-peak data are fitted using at least four parameters (the P's and non-axial recoil parameters?) for H⁺ and five for C₂⁺, so the agreement isn't particularly impressive either.

This leaves me unconvinced that the authors have measured the degree of alignment that they claim.

5. Although indole lacks rotational symmetry, I suspect the effects of this lack of symmetry are negligible as far as this experiment is concerned since the inertia and polarizability tensor axes are very nearly overlapped. Would a model with C_{2v} symmetry not be sufficient to explain the observed alignment? The fact that H⁺ and C₂⁺ from two halves of the molecules are paired up in fitting suggests that it would be. The authors may be overselling this aspect of the paper.

6. I'm surprised to not see any references to Lee et al. (Phys. Rev. Lett. 97, 173001), who first demonstrated field-free 3D alignment, or to Ren et al. (Phys. Rev. Lett. 112, 173602 (2014)), who used measurements techniques similar to those presented here to show that FF3Da can be progressively improved by adding more pulses. The reference to Makhija et al. (ref. 47) is misplaced in the main manuscript. It should be cited where $\cos^2\delta$ is introduced.

In conclusion, this work claims to make important progress but, in my opinion, falls short on delivery. I'll be happy to reconsider my opinion if the authors can address my comments adequately, but I cannot recommend publication in Nature Communications in the current form.

Reviewer #2 (Remarks to the Author):

The paper describes a method for achieving three-dimensional (3D) field-free alignment of general asymmetric-top molecules, and demonstrates it on indole, a rather complex organic molecule. The 3D

alignment is adiabatically induced by a long (psec) shaped elliptically-polarized laser pulse. The pulse is turned off rapidly and the residual alignment persists for several picoseconds. To prove the existence of 3D alignment, the authors provide clear quantitative evidence for the planar alignment, but the in-plane degree of alignment cannot be directly measured by the experiment, and is extracted by theoretical analysis of the observed data under the experimental conditions.

Field-free 3D alignment, and particularly of larger and more complex molecules is an important goal and in this respect the paper addresses a timely and relevant topic. There are, however, several issues that need to be addressed before the paper can be published in NC – some more important than others.

1. The fast turnoff of the shaped pulse is an important asset of the paper. However, in the figure a parasitic field is seen at around 13 psec, and the statement is that during the relevant time window of around 3.3 psec the residual field is less than 1% of the peak value. This is a critical element for the claim of field free alignment. The first question is experimental – how was the 1% value determined, and what is the error level of this estimate? The second is more fundamental – do we know that this residual field does not affect the molecular alignment? It is possible, in principle, that a weak field may not be enough to align the molecule, but might be enough to maintain alignment that had already been achieved. This point cannot be ignored and must be discussed in detail.
2. Another issue is the theoretical analysis of the in-plane degree of alignment. The experimental statement is that

"Furthermore, the degree of molecular alignment retrieved from the angular momentum distributions of H⁺ and C₂⁺ can be misrepresented mainly due to two reasons – the many-body Coulomb break-up of the multiply charged indole cation, with ionic fragments violating the axial-recoil approximation, as well as the indistinguishability of fragments emitted at different molecular sites."

To account for multiple scattering and many body Coulomb breakup, especially for the hydrogen, envelope Gaussian functions are applied to the recoil angle, and are fitted to the measured data. The results are displayed in figure 3, and the claimed degree of in-plane alignment is thus extracted. It seems that there is quite a large degree of uncertainty in such a procedure, the physical nature of the approximation is not intuitive, and a more thorough discussion is needed.

Several other issues require attention:

3. Multiple references in a single citation should be ordered by year of publication. For example, change the order of Refs. 3 and 4, the second one is published in 2007, while the first one in 2013. Additionally, two works [PRL 94, 143002 (2005), "Field-Free Three-Dimensional Molecular Axis Alignment"; PRL 97, 173001 (2006), "Field-Free Three-Dimensional Alignment of Polyatomic Molecules"] should be cited, as they are closely related to this paper. Also, a more recent review on molecular rotation control can be added to [1, 2], i.e. Rev. Mod. Phys. 91, 035005 (2019).

4. Consider citing some of the works on two-color orientation. Although the orientation is in the focus of these works, they are related as the two-color field in a non-parallel configuration induces a 3D alignment.
5. In the second paragraph on page 1, the authors state “the standard approaches to characterize the 3D degree of alignment, using ion imaging of atomic fragments, mostly halogen atoms, recoiling along a well-defined molecular axis, do not work.” A reference should be provided.
6. From the Sup. Inf. “Furthermore, an incoherent average over the initial rotational state distribution, determined from measured experimental deflection profiles, was carried out.” Which rotational states were included? Were they identical to the thermal distribution? If not how different is the considered distribution of initial states from the thermal distribution and why?

7. On page 2, in the left column the experimental system is described:

"The SLM was specifically used for spectral phase modulation of the long wavelength side of the laser spectrum, i. e., components longer than 815 nm. In addition, wavelengths longer than 816 nm were blocked by a razor blade, situated in front of the SLM."

This seems to be inconsistent – Should it be “shorter than 816 nm”?

8. In Fig. 2a, the values of $\langle \cos^2 \theta_{zz} \rangle_1$, $\langle \cos^2 \theta_{yy} \rangle_1$, $\langle \cos^2 \theta_{xx} \rangle_1$ are provided, but these quantities are not defined at this point in the text. What is the value of α in Fig. 2? Also, what is the approximate duration of the kick, ~ 2.5 ps?
9. In the fifth paragraph on page 2, the authors state “After selection of a suitable two-dimensional (2D) radial range, the degree of alignment $\langle \cos^2 \theta_{2D} \rangle$ was computed.” The 2D plane and θ_{2D} should be defined.
10. The statement is that “The strongest field-free alignment was observed near $t = 3.3$ ps.” Why?
11. The isotropic value is given as $\langle \cos^2 \theta_{2D} \rangle_{\text{exp}}^{\text{H}^+} = 0.6$, rather than the expected 0.5. An explanation is required.
12. In the last paragraph on page 3, the authors state “To determine the 3D alignment of indole, an additional observable is required that characterizes the in-plane alignment, i.e., the alignment of the most polarizable axis of indole z_I with respect to the main polarization axis of the alignment field.” Why is it z_I and not y_I ? The alignment of ab -plane ($z_I x_I$ plane) was already discussed in the 2D momentum distributions (Fig. 1).
13. The description of the analysis should be clarified: on page 4, left column “A full tomographic 3D distribution for $\alpha = 0 - 180^\circ$ ($\Delta\alpha = 2^\circ$) was obtained from the measurements of H^+ momentum distributions, see Supplementary Information.” Figure 3 was obtained by rotating the polarization ellipse from $\alpha = 0$ to $\alpha = 180^\circ$, and at each α the total ion count at the center

of the detector was recorded. An explanation is needed as to why the full 3D distribution is essential for producing Fig. 3?

14. A few minor corrections:

- “(x=0,1,2)” in the main text and Fig. 1b should be italic, namely, “(*x* = 0,1,2)”.
- In the last paragraph on page 1, “ $1.25 \cdot 10^{12} \text{ W/cm}^2$ ” should be “ $1.25 \times 10^{12} \text{ W/cm}^2$ ”.
- In the third paragraph on page 4, “incorporate” should likely be “incorporate”.

To summarize, the topic and observations are interesting and timely, but the paper cannot be published until the issues listed above are addressed.

Response to reviewer comments

Reviewer 1

The authors claim a impressive degree of field-free 3D alignment of a somewhat large planar molecule using a long pulse that has been truncated and shaped to kick the molecules into better alignment just as the pulse intensity decays to a negligible level. The demonstration of this idea is an important step forward for the use of aligned molecules in experiments seeking to make “molecular movies”. The molecule indole also has no convenient alignment markers for momentum spectroscopy, so the measurements of 3D alignment is particularly challenging.

Thank you very much for the positive evaluation of our work!

But I have several questions and concerns, which I have listed below.

1. Fig. 2 shows the temporal shape of the pump laser with an intensity scale. How was the peak intensity of the pulse (at 0 ps) estimated? How large is the error bar on this measurement? Was the possible error in the intensity considered in the fitting of the delay-dependent data?? Since the determination of the expectation values of the various squared-cosines relies heavily on simulations, it is important to understand error in the estimated intensity and the effect of the error in the peak intensity.

We have added a new section 'Experimental Details' in the supplementary information (SI I. 57 ff), containing a detailed description of how the laser intensity in the experiment was determined, as well as an error estimate on the intensity.

In brief, the peak intensity was determined by combining measurements of the pulse energy, the temporal profile, and the spatial beam profile. The pulse energy was determined by measuring the average power and dividing by the repetition rate of 1 kHz. The temporal profile of the alignment pulse was determined experimentally by measuring a cross-correlation between the alignment pulse and the Coulomb explosion pulse. Finally, the spatial beam profile was measured on a beam profiler. The estimated statistical error on the measurement is $\approx 7\%$ and the estimated systematic error $< 10\%$.

We note that our simulations confirm that the degree of alignment, in particular the peak alignment observed at 3.3 ps and the revival structure, was not significantly modified when the peak intensity was varied by $\pm 10\%$, in agreement with our error estimates.

2. The discussion of probe-selectivity, or geometric alignment, is limited. Fig. 2 shows that the measured fragment all have significantly aligned momentum distributions without the pump pulse, but there is no mention of geometric alignment in the discussion of the simulations. Is geometric alignment accounted for? If yes, how? If not, why not? Do the simulations accurately reproduce the geometric alignment seen in Fig. 2?

As stated in the manuscript (l. 277–283), the simulations only consider the rotational alignment dynamics induced by the pump laser pulse. However, as mentioned by the referee, geometric alignment must be taken into account in the analysis. The polarisation of the electric field of the probe laser pulse can indeed lead to preferential ionisation directions. In our case, the probe laser was circularly polarised (l. 121–122) leading to an increased ionisation efficiency in a plane perpendicular to our detector for the indole molecule. However, we exactly used circular polarisation, because it minimised the effect of geometric alignment. To quantify the degree of geometric alignment introduced by the probe laser field, ion momentum distributions were recorded without alignment pump laser pulse. For H^+ ions, the measured distribution resulted in a degree of geometric alignment of 0.60 (l. 199–203), see the updated Figure 2 e of the supplementary information. To account for this geometric alignment in our analysis, an additional constant offset was used as a fitting parameter when comparing the results of our measurement to the simulations. The resulting value from the fit was $\cos^2 \hat{O}_{2D} = 0.61$,

practically identical to the measured value of $\cos^2 \theta_{2D} = 0.60$.

There is also no mention of the probe intensity or pulse duration in the manuscript.

The duration of the probe laser was measured to ≈ 35 fs (FWHM) and its peak intensity was estimated to 4.6×10^{14} W/cm². These values were included in the description of the experimental setup in the main manuscript (l. 118–121).

3. The use of fragments near-zero momentum in the detector plane is well-justified for H⁺ fragment through the tomographic measurement of the 3D momentum measurement. But no such measurement is discussed for the C²⁺ fragment. Was such a measurement made?

Tomographic measurements and reconstructions were carried out for the fragments H⁺, C⁺, C²⁺ and CH_xN⁺ (x = 0, 1, 2). The reconstructed three-dimensional velocity distribution of H⁺ ions is shown in Figure 2 of the supplementary information as an example. For both the H⁺ and the C²⁺ fragments, the velocity distributions are quasitoroidal, i. e., there is no considerable density at or around the origin.

We stated in the main manuscript (l. 237–247) that these tomographic measurements were carried out for H⁺, C⁺, C²⁺ and CH_xN⁺ (x = 0, 1, 2), with the 3D momentum distributions for H⁺ and C²⁺ showing negligible density around the origin.

And for H⁺, doesn't the tomography data contain the data that is shown in Fig. 3? Wouldn't a narrow cut through the “doughnut” shown in Fig.3 of SI reproduce the plot for H⁺ shown in Fig. 3? If tomography is necessary to verify that the “masked VMI” measurements are useful – and I think it is – I'm not sure I see the point in using the 2D VMI data directly.

For H⁺ and C²⁺ fragments, the masked-VMI analysis is indeed equivalent to using narrow cuts through the corresponding three-dimensional distributions. However, this equivalence is only valid if the density around the velocity origin is negligible. Otherwise, a reconstruction of the full three-dimensional distribution is indispensable. When valid, our masked VMI analysis substantially decreases the data acquisition time as less ion data per projection angle is required with respect to a full tomographic characterization of the 3D momentum ion distribution. In addition, this approach allows to circumvent the cumbersome task of actually reconstructing the three-dimensional distribution and the necessity of correspondingly high signal-to-noise data. Together, we believe that our masked-VMI analysis is therefore more practical.

We have added a short description of the equivalence of the two methods and the reason for choosing the masked-VMI analysis in the main manuscript (l. 241–257).

Using tomography to characterize 3D alignment isn't a new approach as claimed by the authors. See Ren *et al.* Phys. Rev. Lett. 112, 173602 (2014).

We agree that using tomography for the characterization of the 3D degree of alignment is not a new approach, which we also did not claim at any point in our manuscript. We have included the specified reference (Ren *et al.*, 2014, [36] in the manuscript) (l. 51). What we consider to be new is the combination of two separate measurements for the complete characterization of the planar and in-plane degree of alignment together with the, quite challenging and unusual, determination of the degree of alignment based only on hydrogen and carbon ions, without the need for artificially introduced marker groups. This, we believe, is a key new aspect of our work and a crucial advance.

Visually, the images showing the 3D momentum distributions of H⁺ in the SI is unsatisfactory. The manuscript relies heavily on fitting data to simulations, but the visual representation of the data is such that details of the distribution are hard to see. I suggest the authors consider a better-looking representation, perhaps showing cuts through the distribution in addition to the full 3D distribution.

We have modified Figure 2 in the supplementary information to achieve better visibility of the

reconstructed three-dimensional H⁺ velocity distribution. In the new figure, the isosurface is now shown from a different single perspective, making it easier to see the density at the origin of the velocity coordinate system. Additionally, three slices through the three-dimensional alignment distribution, along three different views of the molecule, are now provided.

4. That brings me to what's my most important concern about this manuscript. The simulation of the momentum distribution seems to have a lot of free parameters to describe data that has little structure. There are the weights for each fragment, the Gaussian function that accounts for non-axial recoil of H⁺ (this function and its parameters are not provided in the SI, although the main article says it is), any parameters needed to account for geometric alignment (if it is accounted for at all – I suspect it is not since the probabilities for various fragments are numbers rather than angle-dependent functions).

In fact, the simulations have very few free parameters (O(10)) compared to the many hundred experimental datapoints.

We have improved the description of the fitting procedure in the supplementary information by adding more details, see subsection 'Comparison of Experiment with Theory' (SI l. 150 ff). In particular, we have included the values of all computed weights together with an estimated error, the offset accounting for the geometric alignment, and the opening angle of the Gaussian function to account for non-axial recoil (SI l. 179). The objective function being minimized was written out in detail and the number of fitted data points and free parameters were stated explicitly (SI l. 173-182). The normalised χ^2 value was also included (SI l. 189). For both, the fit of the revival structure as well as the fit of the angular distributions from the masked-VMI measurements, computed basis functions, extracted from the simulated angular probability density functions, were used for each individual hydrogen and carbon atom, respectively. Comparison of our measured revival dynamics with the theoretical fit results in a normalised χ^2 value of 2.3. Compared to the number of data points used in the fit, on the order of 500 for hydrogen and 90 for carbon, the 8 or 9 fitting parameters that were used result in a reliable fit of the data by our model.

But the pump pulse parameters are not varied. The fitting procedure doesn't really retrieve any alignment parameters – it just serves to confirm that the data agrees with the rigid-rotor TDSE simulations after the simulation data for molecular axis distributions have been suitably fudged. The variational calculation only determines the fudging parameters required to get agreement. Are these parameters reasonable? Is the model used for the fudging appropriate and not unnecessarily large? How are we to judge? On the one hand the data in Fig 2(b) is rich in structure, but the fit is not particularly convincing (and I'd like to see the fit to the data in Fig. 2(a)). On the other, the fit to data in Fig. 3 is good but these two-peak data are fitted using at least four parameters (the P's and non-axial recoil parameters?) for H⁺ and five for C²⁺, so the agreement isn't particularly impressive either.

Part of the raised question was already answered in our reply to question 4, part 1, *vide supra*, in particular the part regarding the fitting procedure and quality of the fit.

The used fitting procedure allowed us to extract the degree of alignment of the molecules from a direct measurement of the H⁺ and C²⁺ ion momentum distributions. To do so, time-dependent rotational probability distributions were computed for every delay time and from them the corresponding probability distributions for every hydrogen and carbon atom of indole individually. The simulations were carried out by using only experimentally determined laser parameters, without any free fitting parameters. The simulations of the rotational dynamics of molecules with the TDSE are believed to correctly describe the alignment dynamics. Our simple Coulomb explosion model and the projections of the individual rotational probability distributions, that were used in the fitting procedure, thus served the purpose of direct comparison with the experimental data. The highly structured alignment revival dynamics observed experimentally in the H⁺ fragments is quite well reproduced by our model. In particular, all oscillations observed

are fully reproduced. In addition, the masked-VMI measurements performed in H^+ and C^{2+} are also fully reproduced. Considering that the molecule doesn't contain any heavy atomic marker and that the H^+ and C^{2+} ion momentum distributions recorded experimentally originate from 7 and 8 distinct localtions in the molecule, respectively, we consider the agreement between our model and the measurement to be quite satisfying.

This leaves me unconvinced that the authors have measured the degree of alignment that they claim.

We hope that our updated discussion, in the supplementary information and above, clarifies how we were able to correctly identify the 'real' degree of alignment of indole in our experiment.

5. Although indole lacks rotational symmetry, I suspect the effects of this lack of symmetry are negligible as far as this experiment is concerned since the inertia and polarisability tensor axes are very nearly overlapped. Would a model with C_{2v} symmetry not be sufficient to explain the observed alignment? The fact that H^+ and C^{2+} from two halves of the molecules are paired up in fitting suggests that it would be. The authors may be overselling this aspect of the paper.

We agree with the referee that the resulting rotational dynamics of indole at the achieved degree of alignment is not significantly affected by the small angle between the inertia and the polarizability frame. However, we point out that this fact was a result of the experiment and the simulations and not a prerequisite we assumed or used as *prior knowledge*. As the referee pointed out, indole itself, i. e., in the molecular frame, is lacking any rotational symmetry, but the rotational wavefunctions of every molecule, i. e., in the laboratory frame, obey Four group symmetry.

We have modified our expression in the manuscript, e. g., in the abstract and l. 53, from 'non - rotation-symmetric molecule' to '(generic) asymmetric top molecule' to avoid any wrong impressions of the reader.

What is crucial for *measuring* the degree of alignment, however, are the recoil vectors of the utilised ionic fragments in the polarisability frame of the molecule. Here, we employed a simple Coulomb-explosion model for the H^+ and C^{2+} ions to link our robust TDSE calculations of the rotational dynamics to the detectable ion distributions. The fact that multiple atoms are grouped together in our Coulomb-explosion model results from *orientational* averaging, which is due to the rotational-wavefunction Four group symmetry, which is always present in molecular alignment experiments, i. e., when orientation effects are absent. For any laser-aligned – but not oriented – molecule, irrespective of its symmetry, the density of its rotational wave function will always possess a symmetry according to the alignment laser's optical-cycle averaged intensity distribution. For the elliptically polarised pulses we used in our experiments, this density would always have D_{2h} symmetry, even for a C_1 (M) symmetric molecular target.

6. I'm surprised to not see any references to Lee et al. (Phys. Rev. Lett. 97, 173001), who first demonstrated field-free 3D alignment, or to Ren et al. (Phys. Rev. Lett. 112, 173602 (2014)), who used measurements techniques similar to those presented here to show that FF3Da can be progressively improved by adding more pulses. The reference to Makhija et al. (ref. 47) is misplaced in the main manuscript. It should be cited where $\cos^2 \theta$ is introduced.

We have included Lee *et al.* (2006, [35] in the manuscript) and Ren *et al.* (2014, [36]) in the introduction. We have also placed the reference to Makhija *et al.* (2012, [51]) to the place in the manuscript where $\cos^2 \theta$ is introduced.

In conclusion, this work claims to make important progress but, in my opinion, falls short on delivery. I'll be happy to reconsider my opinion if the authors can address my comments adequately, but I cannot recommend publication in Nature Communications in the current form.

We believe that we have satisfactorily addressed and clarified all questions and comments raised by this reviewer. We hope that we could convince the reviewer that our method to extract the

degree of alignment for an asymmetric-top molecule like indole, lacking a marker atom to easily quantify the degree of alignment, is an important and crucial step toward 3D field-free alignment and its characterisation for generic asymmetric-top molecules.

Reviewer 2

The paper describes a method for achieving three-dimensional (3D) field-free alignment of general asymmetric-top molecules, and demonstrates it on indole, a rather complex organic molecule. The 3D alignment is adiabatically induced by a long (psec) shaped elliptically-polarized laser pulse. The pulse is turned off rapidly and the residual alignment persists for several picoseconds. To prove the existence of 3D alignment, the authors provide clear quantitative evidence for the planar alignment, but the in-plane degree of alignment cannot be directly measured by the experiment, and is extracted by theoretical analysis of the observed data under the experimental conditions.

Field-free 3D alignment, and particularly of larger and more complex molecules is an important goal and in this respect the paper addresses a timely and relevant topic. There are, however, several issues that need to be addressed before the paper can be published in NC – some more important than others.

Thank you very much for the appropriate summary and the overall very positive evaluation of our work. In the following, we respond to the criticism raised and believe that these issues are resolved with our response and the updated manuscript.

1. The fast turnoff of the shaped pulse is an important asset of the paper. However, in the figure a parasitic field is seen at around 13 psec, and the statement is that during the relevant time window of around 3.3 psec the residual field is less than 1 % of the peak value. This is a critical element for the claim of field free alignment. The first question is experimental – how was the 1 % value determined, and what is the error level of this estimate? The second is more fundamental – do we know that this residual field does not affect the molecular alignment? It is possible, in principle, that a weak field may not be enough to align the molecule, but might be enough to maintain alignment that had already been achieved. This point cannot be ignored and must be discussed in detail.

An extensive discussion of the characterization of the laser pulse parameters, in particular the laser intensity, was already given in our answer to question 1 of reviewer 1, *vide supra*.

We have carried out and compared simulations with and without residual alignment laser fields, set to 0.1 % and 1 % of the peak intensity of the alignment laser pulse; the results are shown in the SI, new Figure 5. As one can clearly see in this Figure the effect of a residual field on the order of 1 % of the peak intensity does not have any influence on the degree of alignment and its temporal evolution up to 13 ps, in particular the value of the maximum degree of alignment is the same in all cases.

We have added a paragraph in the supplementary information (l. 226-239 in SI), discussing the results from the simulations shown in Figure 5 of the SI and stating that a possible residual field on the order of up to 1 % does not have any effect on the measured degree of alignment in the region of interest, which extends up to 13 ps, where an unwanted postpulse emerges.

2. Another issue is the theoretical analysis of the in-plane degree of alignment. The experimental statement is that “Furthermore, the degree of molecular alignment retrieved from the angular momentum distributions of H⁺ and C²⁺ can be misrepresented mainly due to two reasons – the many-body Coulomb break-up of the multiply charged indole cation, with ionic fragments violating the axial-recoil approximation, as well as the indistinguishability of fragments emitted at different molecular sites.”

To account for multiple scattering and many body Coulomb breakup, especially for the hydrogen,

envelope Gaussian functions are applied to the recoil angle, and are fitted to the measured data. The results are displayed in figure 3, and the claimed degree of in-plane alignment is thus extracted. It seems that there is quite a large degree of uncertainty in such a procedure, the physical nature of the approximation is not intuitive, and a more thorough discussion is needed.

A more detailed description of the fitting procedure is now provided in the section “Comparison of Experiment with Theory” in the supplementary information (SI l. 150); see also our answer to question 4 of reviewer 1.

The TDSE simulations were performed using experimentally determined parameters for the laser field and molecular parameters obtained either from literature or from elaborate quantum chemistry simulations: There was no fitting parameter involved in this step. Following that, rotational probability distributions were computed for every hydrogen and carbon atom of indole separately. In order to verify that the simulations provide the correct rotational probability distributions, we computed the observables that were experimentally measured and directly compared these to the experimental results. For this, we employed a simple Coulomb explosion model, in which the contribution of every hydrogen and carbon atom to the overall observables was computed. The angular distribution of the rotational probability functions was thus fixed and not subject to any fitting. The computed observables fitted extremely well the C^{2+} results, however a discrepancy in the in-plane alignment for H^+ was observed. We attributed this to the possibility of non-axial recoil of the singly charged hydrogen atoms and modelled this by a convolution of the angular distributions with a Gaussian function, thus smearing out the angular distributions and leading to broader distributions in the final momenta of the observed H^+ fragments Christensen *et al.* (2016). We believe, on physical grounds, that this is a good and feasible approximation to account for non-axial recoil in a simple manner.

Several other issues require attention:

3. Multiple references in a single citation should be ordered by year of publication. For example, change the order of Refs. 3 and 4, the second one is published in 2007, while the first one in 2013. Additionally, two works [PRL 94, 143002 (2005), “Field-Free Three-Dimensional Molecular Axis Alignment”; PRL 97, 173001 (2006), “Field-Free Three-Dimensional Alignment of Polyatomic Molecules”] should be cited, as they are closely related to this paper. Also, a more recent review on molecular rotation control can be added to [1, 2], i.e. Rev. Mod. Phys. 91, 035005 (2019).

We have added the more recent review article by Koch *et al.* (2019, [3]). We have also added Underwood *et al.* (2005, [34]) and Lee *et al.* (2006, [35]). Furthermore, we have put all multiple references throughout the manuscript into chronological order.

4. Consider citing some of the works on two-color orientation. Although the orientation is in the focus of these works, they are related as the two-color field in a non-parallel configuration induces a 3D alignment.

There is a vast bibliography available on alignment and orientation of molecules in the gas phase, including work on orientation by ourselves. However, we believe that we have included all relevant references for our work in the manuscript and we consider that such citations focused on orientation are beyond the scope of our research article.

5. In the second paragraph on page 1, the authors state “the standard approaches to characterize the 3D degree of alignment, using ion imaging of atomic fragments, mostly halogen atoms, recoiling along a well-defined molecular axis, do not work.” A reference should be provided.

We can't provide a specific reference, as we believe this statement as such will not be found in the existing literature – i. e., we considered to add such a reference ourselves, but do not know of any appropriate publication. However, when evaluating the literature, one recognises that almost all work on the alignment of complex asymmetric top rotors was carried out on molecules which contain clear, heavy marker atoms, or sometimes pseudohalogens such as CN (Holmegaard *et al.*, 2010; Hansen *et al.*, 2013), whose recoil defines the principal inertia and polarisability

axes in these molecules well. Furthermore, typically molecules are used that contain only one or very few of these marker atoms, which aids in temporal gating of detectors to measure these fragments isolated from other fragments, enabling a straightforward characterisation of the degree of alignment.

We do not have the same situation in our work as indole lacks any such “marker” atoms, consisting only of hydrogen, carbon and nitrogen – and even with all C and N atoms being part of rings, i. e., requiring the breaking of at least two bonds.

6. From the Sup. Inf. “Furthermore, an incoherent average over the initial rotational state distribution, determined from measured experimental deflection profiles, was carried out.” Which rotational states were included? Were they identical to the thermal distribution? If not how different is the considered distribution of initial states from the thermal distribution and why?

The measured deflected experimental distribution of rotational states in the interaction region that was used for the incoherent average over initial states differs from a thermal Boltzmann distribution. Rotational states are spatially dispersed by the inhomogeneous electrostatic field inside the deflector according to their effective dipole moment. The effect is larger for the rotational ground state as it has the largest effective dipole moment. This was previously described in detail by Filsinger *et al.* (2009) and referenced in the main manuscript (see lines 77-78).

The total population for every J-state, summed over all (K,M)-substates is shown in a new Fig 3 in the supplementary information. This shows the clear difference between the actual distribution and a thermal distribution at 1 K, e. g., in a direct molecular beam. We have also added a brief description in the supplementary information (SI I. 144-148) stating the difference between a thermal Boltzmann distribution and the used deflected-states distribution.

7. On page 2, in the left column the experimental system is described: “The SLM was specifically used for spectral phase modulation of the long wavelength side of the laser spectrum, i. e., components longer than 815 nm. In addition, wavelengths longer than 816 nm were blocked by a razor blade, situated in front of the SLM.” This seems to be inconsistent – Should it be “shorter than 816 nm”?

There were two devices modifying the pulse spectrum in the Fourier plane of the 4f shaper setup: the SLM modified the phase for wavelengths > 815 nm and a knife edge independently modified (blocked) the amplitudes of wavelengths > 816 nm. The manuscript was modified to clarify this point (l. 101-103).

8. In Fig. 2a, the values of $\cos^2 \alpha_{ZZI}$, $\cos^2 \alpha_{YYI}$, $\cos^2 \alpha_{XXI}$ are provided, but these quantities are not defined at this point in the text. What is the value of α in Fig. 2? Also, what is the approximate duration of the kick, ≈ 2.5 ps?

We have now included a description of α and its meaning in the text, referring to Figure 1 and Figure 2, see the caption of Figure 1 and lines 136-144 in main manuscript. We have added the value of $\alpha = 0^\circ$ in the caption of Figure 2, as well as a description of the expectation values and the angles/indices. The duration of the kick before truncation is 2.6 ps (FWHM). We have included this value in the description of the experimental setup in the main manuscript (l. 98 ff).

9. In the fifth paragraph on page 2, the authors state “After selection of a suitable two-dimensional (2D) radial range, the degree of alignment $\cos^2 \alpha_{2D}$ was computed.” The 2D plane and α_{2D} should be defined.

We have included a brief description of the imaging geometry in the main manuscript (l. 137–145) where the experimental setup and the extraction of the observables from the data are discussed, which now also defines the 2D detector plane and the polar angle α_{2D} within this plane.

0. The statement is that “The strongest field-free alignment was observed near $t = 3.3$ ps.”

Why?

The mechanism leading to field-free alignment of molecules consists in the creation of a coherent superposition of rotational eigenstates of the molecule. The created wavepacket depends thereby critically on several parameters. First of all it depends on the rotational spectrum of the molecule, i. e., the energy level structure, which in turn depends on the moments of inertia of the molecule and hence its structure. Furthermore, the alignment dynamics depends on the exact parameters of the laser pulse (duration, shape, polarisation, intensity). Generally, strong alignment is achieved when all populated states in the wavepacket are in phase. The degree of field-free alignment is determined by the populations of these states and their phase-spread (Karamatskos *et al.*, 2019, Fig. 3).

It takes some time for the phases of the involved states to evolve such that their phases match and alignment is achieved. This is generally referred to as the “prompt alignment”, which in all field-free alignment scenarios using short laser pulses, much shorter than the rotational motion of the molecule, occurs, showing the peak alignment some time after the interaction of the molecules with the alignment laser pulse. Classically, one could imagine that the molecular axes are randomly, isotropically distributed in space and through the interaction with the laser electric field a torque is exerted such that all molecules start rotating towards the instantaneous laser alignment polarisation axis. Since the molecules have inertia, this takes some time, which is the 3.3 ps observed in our experiment.

Note that depending on the shape of the laser pulse the prompt alignment can be higher or lower than the peak adiabatic alignment in the field. In simulations, performed prior to the actual experiment, we experimented with different possible pulse shapes, which we could achieve with our pulse shaper, and selected the best candidate, showing the highest degree of alignment under field-free conditions, and not at the peak of the adiabatic ramp; see also Rouzee *et al.* (2009).

11. The isotropic value is given as $\langle \cos^2 \theta \rangle_{\text{exp}} = 0.6$, rather than the expected 0.5. An explanation is required.

A similar question was raised by reviewer 1. Please see our answer to question 2 of referee 1, *vide supra*.

12. In the last paragraph on page 3, the authors state “To determine the 3D alignment of indole, an additional observable is required that characterizes the in-plane alignment, i. e., the alignment of the most polarisable axis of indole z_I with respect to the main polarisation axis of the alignment field.” Why is it z_I and not y_I ? The alignment of ab-plane (z_I/x_I plane) was already discussed in the 2D momentum distributions (Fig. 1).

The (a, b) plane coincides with the (z_I, x_I) plane, whereas c and y_I are perpendicular to the molecular plane. Alignment of the plane (z_I, x_I) fixes also the axis y_I within the rigid rotor approximation, since they are always perpendicular. However, the degree of alignment of the plane still leaves the question about the orientation of the z_I axis within that plane. A small rotation about the y_I axis will not lead to a large change in the degree of alignment of the plane, hence this is not sufficient information to determine the degree of alignment of the z_I axis. Thus, instead we derive the alignment of z_I , which is also the most polarisable axis of the molecule.

13. The description of the analysis should be clarified: on page 4, left column “A full tomographic 3D distribution for $\alpha = 0-180^\circ$ ($z\alpha = 2^\circ$) was obtained from the measurements of H^+ momentum distributions, see Supplementary Information.” Figure 3 was obtained by rotating the polarisation ellipse from $\alpha = 0$ to $\alpha = 180^\circ$, and at each α the total ion count at the center of the detector was recorded. An explanation is needed as to why the full 3D distribution is essential for producing Fig. 3?

A similar question was raised by reviewer 1. The answer to this question is contained in our answer to question 3 of reviewer 1, *vide supra*.

14. A few minor corrections:

- “(x=0,1,2)” in the main text and Fig. 1b should be italic, namely, $\mathbf{X} = 0, 1, 2$.
- In the last paragraph on page 1, $1.25 \cdot 10^{12}$ W/cm² should be $1.25 \xi 10^{12}$ W/cm².
- In the third paragraph on page 4, “incorporate” should likely be “incorporate”.

We have fixed the aforementioned typos and stylistic inconsistencies in the manuscript and in Fig. 1.

To summarize, the topic and observations are interesting and timely, but the paper cannot be published until the issues listed above are addressed.

We thank the reviewer for the positive evaluation of our work and the useful comments that we have all addressed appropriately. We have included all discussed changes in the main manuscript and the supplementary information and we believe that the paper is now much stronger. We hope that after these changes and improvements, the manuscript will be considered for publication in Nature Communications.

References

- X. Ren, V. Makhija, and V. Kumarappan, Multipulse three-dimensional alignment of asymmetric top molecules, Phys. Rev. Lett. **112**, 173602 (2014).
- K. F. Lee, D. M. Villeneuve, P. B. Corkum, A. Stolow, and J. G. Underwood, Field-free three-dimensional alignment of polyatomic molecules, Phys. Rev. Lett. **97**, 173001 (2006).
- V. Makhija, X. Ren, and V. Kumarappan, Metric for three-dimensional alignment of molecules, Phys. Rev. A **85**, 033425 (2012).
- L. Christensen, L. Christiansen, B. Shepperson, and H. Stapelfeldt, Deconvoluting nonaxial recoil in Coulomb explosion measurements of molecular axis alignment, Phys. Rev. A **94**, 023410 (2016).
- C. P. Koch, M. Lemeshko, and D. Sugny, Quantum control of molecular rotation, Rev. Mod. Phys. **91**, 035005 (2019).
- J. Underwood, B. Sussman, and A. Stolow, Field-free three dimensional molecular axis alignment, Phys. Rev. Lett. **94**, 143002 (2005).
- L. Holmegaard, J. L. Hansen, L. Kalthøj, S. L. Kragh, H. Stapelfeldt, F. Filsinger, J. Küpper, G. Meijer, D. Dimitrovski, M. Abu-samha, C. P. J. Martiny, and L. B. Madsen, Photoelectron angular distributions from strong-field ionization of oriented molecules, Nat. Phys. **6**, 428 (2010), arXiv:1003.4634 [physics].
- J. L. Hansen, J. J. Omiste, J. H. Nielsen, D. Pentlehner, J. Küpper, R. González-Férez, and H. Stapelfeldt, Mixed-field orientation of molecules without rotational symmetry, J. Chem. Phys. **139**, 234313 (2013), arXiv:1308.1216 [physics].
- F. Filsinger, J. Küpper, G. Meijer, L. Holmegaard, J. H. Nielsen, I. Nevo, J. L. Hansen, and H. Stapelfeldt, Quantum-state selection, alignment, and orientation of large molecules using static electric and laser fields, J. Chem. Phys. **131**, 064309 (2009), arXiv:0903.5413 [physics].
- E. T. Karamatskos, S. Raabe, T. Mullins, A. Trabattoni, P. Stammer, G. Goldsztejn, R. R. Johansen, K. Długołcki, H. Stapelfeldt, M. J. J. Vrakking, S. Trippel, A. Rouzée, and J. Küpper, Molecular movie of ultrafast coherent rotational dynamics of OCS, Nat. Commun. **10**, 3364 (2019), arXiv:1807.01034 [physics].
- A. Rouzee, A. Gijbetsen, O. Ghafur, O. M. Shir, T. Baeck, S. Stolte, and M. J. J. Vrakking, Optimization of laser field-free orientation of a state-selected no molecular sample, New J. Phys. **11**, 105040 (2009).

REVIEWER COMMENTS

Reviewer #1 (Remarks to the Author):

The authors have addressed most of my concerns satisfactorily. Most importantly, the model used for fitting is now clearly described and my question about the number of fitting parameters is resolved.

But I still take issue with the claim that the authors have “extracted” or “characterized” the degree of alignment from the momentum imaging data. If, as the authors state, it is accepted that the TDSE simulation correctly describes the rotational dynamics (and I agree with this statement), then all that is left to be done is to make sure all the input parameters (initial state distribution, pump pulse properties, polarizability and moment of inertia tensors) required for the TDSE are determined accurately enough and the degree of alignment achieved then follows from the TDSE results. The authors don't really extract the alignment from the momentum data – they do not vary the input parameters for the TDSE based on the momentum measurements, and therefore the momentum distributions had no influence on the value of $\cos^2\delta$ they report. The degree of alignment is not even a parameter in Eq. 3 of the SM – the simulated $\cos^2\theta_{2D}(t)$ values are fixed.

The language in the whole article should be changed to avoid claiming that the degree of alignment is obtained/extracted from the momentum distributions. At best, the method used can only find the momentum distributions to be consistent with the TDSE calculations.

There are no error bars on the data in Fig 2(b). That makes it difficult to judge the quality of the fit.

I must also point out that the alignment of asymmetric top molecules without marker atoms, using fits to the delay-dependent signals, is not a novel idea. See Makhija et al., arXiv:1611.06476, where the degree of alignment of ethylene was determined without fragmenting the molecule at all, while simultaneously characterizing geometric alignment by the probe. The method used there is quite general and applies, in principle, to orientation and 3D alignment as well. The authors should cite relevant literature on the use of delay-dependence to obtain the degree of alignment to provide context to their approach.

Once these issues are addressed, I can recommend publication.

Reviewer #2 (Remarks to the Author):

The authors have addressed the issues raised in the original review and have expanded some of the explanations. Together with the extended Supplementary Information, the experimental situation is much clearer, and my concerns were answered.

While the paper is still complex and not easy to read or comprehend, I think that its present, improved version can now be published in Nature Communications.

Response to reviewer comments

Reviewer 1

The authors have addressed most of my concerns satisfactorily. Most importantly, the model used for fitting is now clearly described and my question about the number of fitting parameters is resolved.

Thanks

But I still take issue with the claim that the authors have “extracted” or “characterized” the degree of alignment from the momentum imaging data.

We updated the manuscript throughout to fulfil the language changes suggested below “to avoid claiming that the degree of alignment is obtained/extracted from the momentum distributions”, for details please see the PDF with changes marked.

If, as the authors state, it is accepted that the TDSE simulation correctly describes the rotational dynamics (and I agree with this statement), then all that is left to be done is to make sure all the input parameters (initial state distribution, pump pulse properties, polarizability and moment of inertia tensors) required for the TDSE are determined accurately enough and the degree of alignment achieved then follows from the TDSE results. The authors don't really extract the alignment from the momentum data – they do not vary the input parameters for the TDSE based on the momentum measurements, and therefore the momentum distributions had no influence on the value of $\cos^2 \Delta$ they report. The degree of alignment is not even a parameter in Eq. 3 of the SM – the simulated $\cos^2 \Delta(t)$ values are fixed.

We added two sentences to the manuscript (p. 5, l. 316 ff) to point out the general option to vary input parameters to the simulations – and that this was not necessary in the current work as we had determined these accurately in our experiments.

The language in the whole article should be changed to avoid claiming that the degree of alignment is obtained/extracted from the momentum distributions. At best, the method used can only find the momentum distributions to be consistent with the TDSE calculations.

vide supra

There are no error bars on the data in Fig 2(b). That makes it difficult to judge the quality of the fit.

We have updated the caption of Fig. 2 in the manuscript to clearly state that the line-thickness of the experimental data corresponds to the experimental standard error of the data.

I must also point out that the alignment of asymmetric top molecules without marker atoms, using fits to the delay-dependent signals, is not a novel idea. See Makhija et al., arXiv:1611.06476, where the degree of alignment of ethylene was determined without fragmenting the molecule at all, while simultaneously characterizing geometric alignment by the probe. The method used there is quite general and applies, in principle, to orientation and 3D alignment as well. The authors should cite relevant literature on the use of delay-dependence to obtain the degree of alignment to provide context to their approach. Once these issues are addressed, I can recommend publication.

We updated the manuscript to include a references on the time-dependent analysis of alignment of asymmetric tops (p. 1, l.42). However, the further specified preprint by Makhija et al. (2016), does not seem to be appropriate as a reference to us, likely in line with this five-year old preprint not being peer-review published.

Reviewer 2

The authors have addressed the issues raised in the original review and have expanded some of the explanations. Together with the extended Supplementary Information, the experimental situation is much clearer, and my concerns were answered.

While the paper is still complex and not easy to read or comprehend, I think that its present, improved version can now be published in Nature Communications.

Thank you.

References

V. Makhija, X. Ren, D. Gockel, A.-T. Le, and V. Kumarappan, Orientation resolution through rotational coherence spectroscopy, preprint (2016), [arXiv:1611.06476](https://arxiv.org/abs/1611.06476) [physics].

REVIEWERS' COMMENTS

Reviewer #1 (Remarks to the Author):

My concerns have been adequately addressed, and I'm happy to recommend publication.